# Evaluating user experience with immersive technology in simulation-based education: A modified Delphi study with qualitative analysis

**Chris Jacobs**[1]*, **Georgia Foote**[1], **Michael Williams**[2]

**1** Department for health, University of Bath, Bath, United Kingdom, **2** School of Medicine, Dentistry and Biomedical Sciences, Queen's University of Belfast, Belfast, United Kingdom

* cj511@bath.ac.uk

## Abstract

### Background

Immersive technology is becoming more widespread in simulation-based medical education with applications that both supplement and replace traditional teaching methods. There is a lack of validated measures that capture user experience to inform of the technology utility. We aimed to establish a consensus of items and domains that different simulation experts would include in a measure for immersive technology use.

### Methods

A 3-stage modified Delphi using online software was conducted to support the conceptual framework for the proposed measure. The first round was informed by prior work on immersive technology in simulation. In the first round, participants were asked to describe what we could measure in simulation-based education and technology. Thematic analysis generated key themes that were presented to the participants in the second round. Ranking of importance in round 2 was determined by mean rank scores. The final round was an online meeting for final consensus discussion and most important domains by experts were considered.

### Results

A total of 16 simulation experts participated in the study. A consensus was reached on the ideal measure in immersive technology simulation that would be a user questionnaire and domains of interest would be: what was learnt, the degree of immersion experienced, fidelity provided, debrief, psychological safety and patient safety. No consensus was reached with the barriers that this technology introduces in education.

### Conclusions

There is varied opinion on what we should prioritise in measuring the experience in simulation practice. Importantly, this study identified key areas that aids our understanding on how

**Data Availability Statement:** All relevant data are within the paper and its Supporting Information files.

**Funding:** CJ received a grant from Simulation West Network as part of Health Education England 2022 funding. No number is associated with grant and no sponsor involvement took place.

**Competing interests:** The authors have declared that no competing interests exist.

we can measure new technology in educational settings. Synthesising these results in to a multidomain instrument requires a systematic approach to testing in future research.

## Background

Teaching through simulation is well established in medical disciplines [1]. The pedagogy underlying simulation is based on deliberate practice through recreation of clinical scenarios, irrespective of the enabling technology. An educational experience is constructed with elements that resemble relevant environments, with a greater or lesser degree of fidelity, and match the functional task with the learner's engagement [2]. For example, we might describe a high-fidelity manikin and clinically accurate context, for training for students in basic life support [3].

In recent decades there are examples of technological innovation enhancing educational effectiveness through increased immersion, associated with improved motivational states [4]. There are many technological modalities of simulation, summarised in Table 1.

The Simulation Process provides an overarching structure to simulation-based education (SBE) (Fig 1). Immersive technology integration in SBE requires consideration of all levels of the process, as no technological modality is inherently superior, and each possesses different strengths and weaknesses relevant for different scenarios.

The reality-virtuality continuum describes a range of simulation environments, from real world to virtual world. Within the continuum, the sensory stimuli can be a mix of real and virtual objects. For example, Mixed Reality (MR) is an overarching heading that includes augmentation of real-life (AR), and virtual environments in a head-mounted display (VR) [6].

The processing abilities of newer VR headsets and rendering of visual content has enabled for high fidelity visual representations of immersive educational experiences, for example, as Fig 2 demonstrates the first-person perspective of a student learning management of sepsis in the acute hospital environment. The extent of real-world disconnect and suspension of disbelief is dependent on the technology. Coherence reflects the authenticity or fidelity with which the technology matches to the real-world, and again there is a spectrum of coherence [7].

There is ever growing evidence that immersive technology benefits healthcare students and qualified health professionals in simulation practice [4, 8–10]. Studies investigating the effectiveness of differing technologies in varied medical specialties is increasing year on year, with research in both VR and AR doubling in the period of 2020–2021 [11].

In a systematic review on immersive technology in healthcare education, over 50 methods were described of measuring healthcare practice, which could be broadly separated into:

**Table 1. Simulation modalities with mixed reality methods.** AR- Augmented reality, VR- Virtual Reality, 360–360-degree video, CAVE- Cave Automatic Virtual Environments, VP- Virtual patient. Modified from Forrest and McKimm [5].

| Simulation modality | Mixed reality modalities |
|---|---|
| Simulated patient | AR, VR, 360, CAVE, VP |
| Part task trainer | AR, VR, 360, and CAVE |
| Virtual learning environment | VP |
| Manikin | AR, VR, 360 |
| In-situ simulation | AR, VR, 360, and CAVE |
| Role play | AR, VR, 360, CAVE, VP |

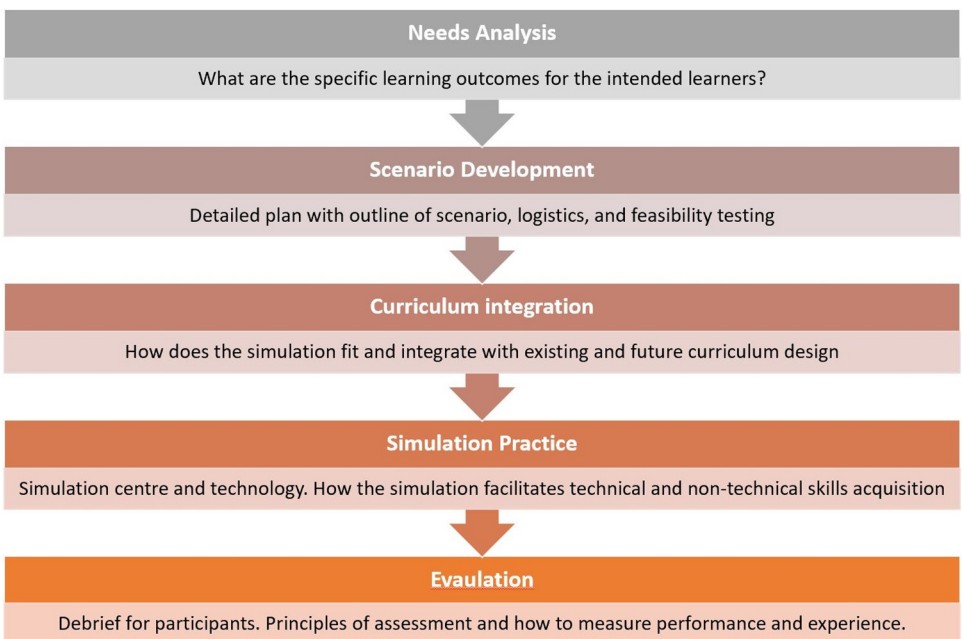

**Fig 1. Simulation process design in simulation-based education.**

cognitive objective, procedural skills objective, or subjective behavioural. Capturing this information helps educators judge the performance of participants and the utility of the technology. The quality of the assessment tool used is essential for evaluating how potentially generalisable the results are [12, 13]. However, shortcomings in the validity and evidence supporting the use of many assessment tools in SBE exist [13, 14].

The Delphi technique is widely accepted as an effective group-based exercise, which aims to integrate multiple interpretations and opinions on a given topic [15, 16]. It has 4 defining characteristics: group anonymity, repeating questions (iteration), feedback to participants, and statistical description of responses [17]. The Delphi researchers orchestrate focal issues surrounding the mode of delivery, the threshold for consensus and encouragement of responses [15]. Delphi techniques supplement analyses of published literature on a topic in preparing a comprehensive framework. Expert consensus in priorities of simulation research has been conducted in several capacities. In a Delphi to develop a cohesive research framework for surgical simulation identified those of highest importance to surgeons: does simulation assessment discriminate performance levels and can competency be translated beyond the simulation [18]. In a separate study, nursing research priorities to a build more rigorous simulation research based on committee feedback highlighted the need for evaluation tools for learner assessment that can be evidenced by validity and reliability to investigate the types of feedback [19]. An established education authority in England (Health Education England), which provides clinicians with simulation aims and objectives, detailed a national strategy for simulation and immersive technologies in health care [20]. The national framework recognised the need to strengthen education-based research in technology enhanced learning. In particular, to better understand the affordances of new technologies. Simulation in the various modalities offers learning in many clinical settings that can be applied MR technology [21].

To date there has not been a consensus report to establish how the research priorities in measuring outcomes can be addressed in immersive technology in healthcare education.

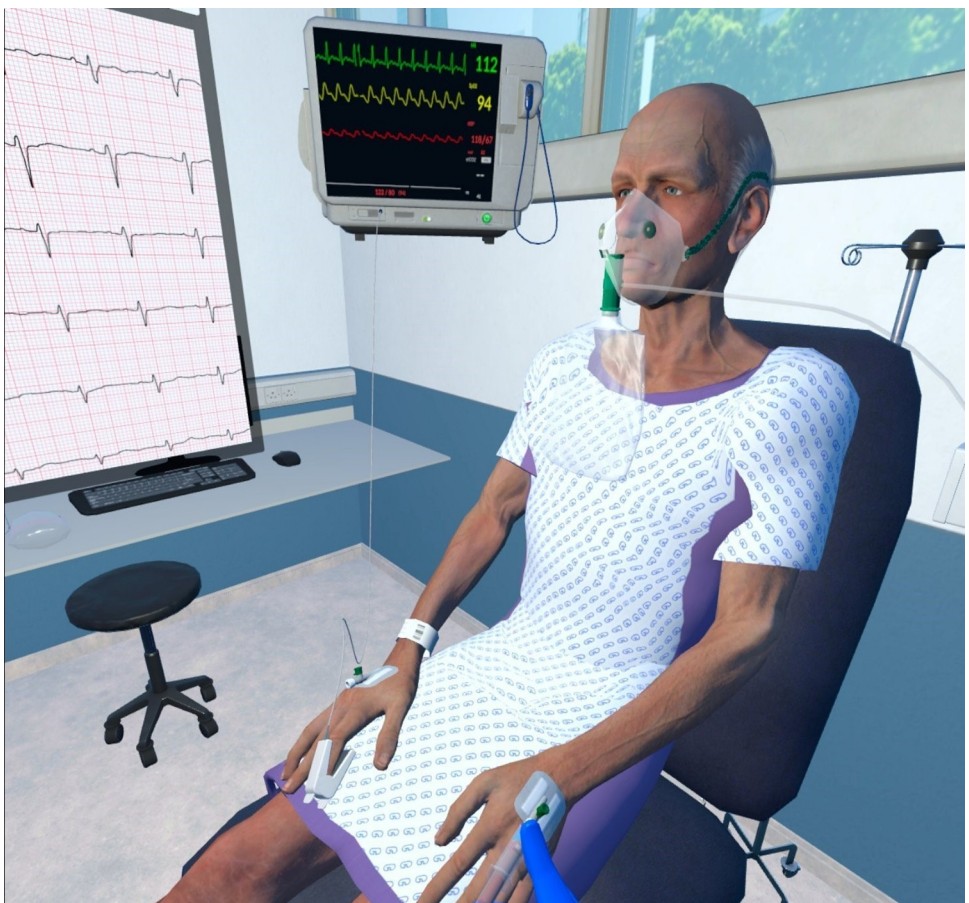

**Fig 2. Virtual reality (VR) interactive experience of a deteriorating patient with sepsis in the emergency room.**
Republished from Goggleminds Ltd under a CC BY license, with permission from Goggleminds Ltd original copyright [2023].

The overarching aim of this project is to create a *de* novo brief, valid and reliable multi-dimensional measure that will inform educators and learners on the users' experiences of technology in simulation practice. The user is situated and operating in a subjective position in simulation and as a participant they are active with agency and have an intentional relationship with MR environment [22]. Technology mediated experiences bring capabilities whereby the subjectivity of the user is framed in an objective sensorimotor interaction that that is a technological driven experience [23].

The domains that could be considered important in quantifying a user's experience are yet not fully understood in simulation. The Delphi approach allows for a diverse number of experts in the field of SBE to be brought together to address this.

## Materials and methods

This study will establish a consensus for generating a new measure's theoretical and conceptual framework through a refining two round Delphi survey and a stakeholder consensus meeting. The Delphi study was approved by Health Research Authority (research ethics committee reference 22/PR/0339 and integrated research application system project ID 312830). Research sponsor was the Great Western Hospital NHS Foundation Trust (UK). Written consent was

obtained following participant information within an embedded online survey for rounds 1 and 2. Round 3 consent was verbally recorded during the video conferenced meeting.

## Modified Delphi method

A steering group consisting of authors (CJ, GF, and MW) guided three-stages of convergent opinion and consensus building [24] between May and August 2022. The conventional Delphi method uses iterative rounds of data collection from groups of experts, with subsequent feedback on responses, with the opportunity for them to rethink their response until consensus is met [25]. Online questionnaire collection facilitates this process for response collection and feedback, in short time frames [26].

A hybrid of the conventional method, a modified Delphi method was adopted, which collects initial participant opinions online, followed by in-person group meetings for final consensus discussion [15, 27]. This was to establish a group-think approach to the final prioritisation of data in earlier rounds, which can benefit from an in person discussion and review of field notes [28]. An *a priori* response rate was set at 70% for Delphi round 2, which was deemed as minimally acceptable to minimise bias [29]. A lower response to the online meeting was deemed acceptable as it required greater commitment, and drop-out linked to additional demands of attending [30]. Fig 3 flow chart illustrates the Delphi process, and a reporting checklist is provided in supplementary material.

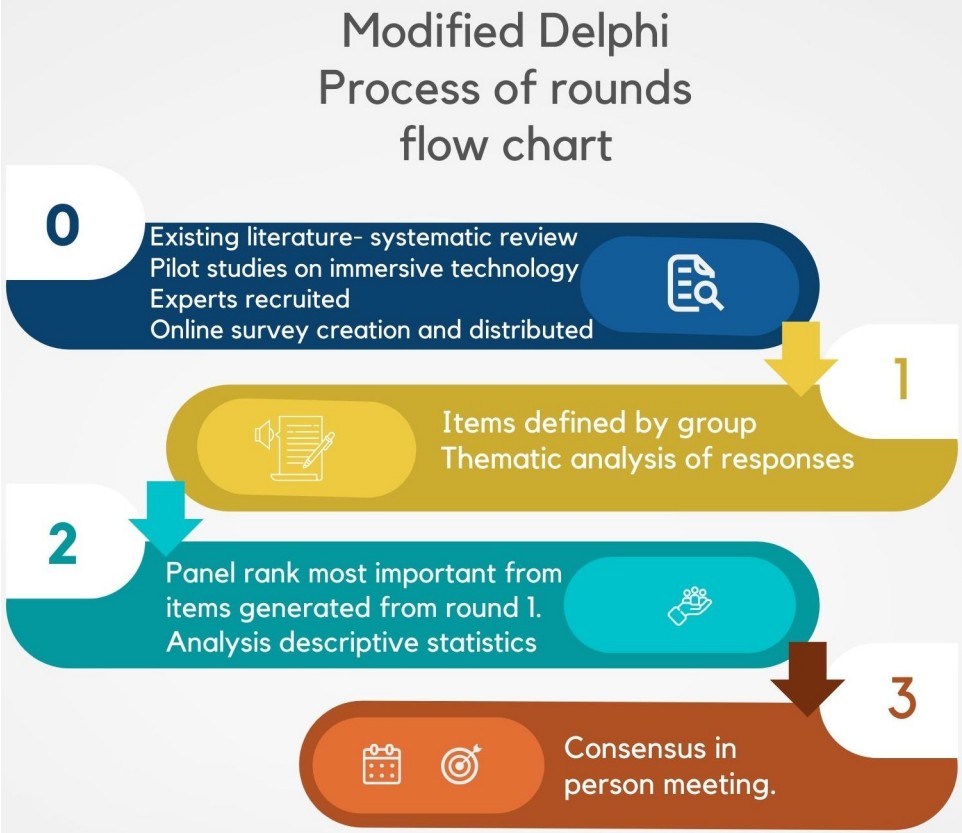

**Fig 3. Evaluating user experience with immersive technology in simulation-based education modified Delphi process flow chart.**

**Table 2. Eligibility criteria- key inclusion and exclusion criteria.**

| Key inclusion criteria |
| --- |
| Experience with simulation in healthcare education setting in the Southwest of England |
| Able to access the internet (Study sign-up and intervention could only be accessed through the internet). |
| Willing to participate in 2 rounds of questionnaires AND/OR willing participate in a virtual meeting. |

## Participants

The target population for this study will be individuals engaged in SBE. Stake holders included doctors, technicians, managers, and administrators involved in simulation for healthcare education (Table 2). Purposive sampling from a targeted group allowed creation of a group of 16 experts, collectively offering a range of experience in healthcare simulation. It was the aim to recruit 10 to 20, which has been described as the panel size to allow meaningful statistical analysis [26, 31]. Recruitment was via a Southwest simulation network email list as a single advert.

## Data collection and analysis

### Delphi round 1

This was an online questionnaire. Questionnaire design and distribution was done using Jisc Online surveys (https://www.onlinesurveys.ac.uk accesse September 2022), software which meets data protection regulations. Participants' information and consent were embedded in the survey prior to questions.

Participants' age and roles were collected to inform authors of range of their simulation experience. Otherwise, question development was drawn from 2 sources: firstly, a qualitative study exploring themes of participant experience in Virtual Reality within an education setting [32], and secondly, a systematic review was undertaken on immersive technology in healthcare education covering literature from 2002 and 2022 [11]. In the systematic review, data from 246 papers on learning theory and on measures adopted in simulation practice with immersive technology were analysed. The instruments that were used by researchers in MR in healthcare education were broadly defined under Blooms taxonomy; cognitive assessment; psychomotor or procedural assessment; and behavioural or affective assessment [33]. Furthermore, literature was appraised using the medical education research study quality (MERSQI) and evaluations found VR based research as the higher methodological quality of all studies in the systematic review. Outcomes from both these sources were discussed between authors CJ and MW to inform questionnaire design and participant information sheets, which briefly summarised key points from the literature.The distributed questionnaire had focused on the modalities of learning in SBE and this instrument also, encouraged participants to write detailed feedback on the following questions: list up to 5 factors that you think are important in learning in simulation, what you think should be measured, what measurements could you use, and what are the barriers you consider in "measuring immersive technology" in simulation and medical education. The authors aimed to explore the experts' opinions and experiences in measuring outcomes in medical education and how this might be applied to immersive technology. Additionally, these research questions overlapped those considered in the literature review and prior pilots in immersive technology to provide the authors with triangulation of data for development of an instrument. Survey length of limited topic-based questions was structure on 4 questions to ensure participant engagement [34].

Round 1 provided qualitative data: panelists responded with free text answers in the form of a non-prioritised list. Thematic analysis was conducted to these open-ended questions with a 6-step process, which supports the integration of surveys in the mechanism of data collection

[35, 36]. Inductive reasoning guided the analysis of responses to identify key patterns and relationships. Generation of initial codes was done independently by CJ and GF. Further interpretative analysis was undertaken by CJ and GF, specifically repeated conversations and code sorting to identify overarching themes, that were eventually defined in short phrases or words that unified the narrative of responses. A thematic map was created as a pictorial summary for feedback to the expert panelists, as well as, the full codes from round 1.

## Delphi round 2

This was an online questionnaire hosted on JISC online surveys. Those participating in round 1 were invited to respond to round 2, which included the same questions with the newly identified and defined key themes resulting from round 1 as options for ranking items in importance from highest to lowest using a check-list response. Means, median and interquartile range (IQR) were calculated using StatsDirect version 3.3.5 for the ordinal round 2 responses [31]. Statistical significance of data was determined as $p < 0.05$, assessed using non-parametric Friedman's test and weighted Kappa to describe strength of agreement between experts ranking [37]. Kappa values were treated as 0.00 to 0.10 conferring poor agreement, 0.11 to 0.20 conferring slight agreement, 0.21 to 0.40 conferring fair agreement, 0.41 to 0.60 conferring moderate agreement, and 0.61 to 0.80 conferring substantial agreement [38].

Mean rankings were complemented with a boxplot visual summary for each question [15], and overarching thematic maps from Round 1 were distributed to participants prior to round 3.

## Delphi round 3

The online meeting was held using Zoom cloud-based videoconferencing service [39]. A secure collaborative online meeting was chosen to facilitate access with participants over a large geographic area [40]. Prior to the scheduled meeting participants received their invitation via email, which included a passcode. The meeting was chaired by CJ and field notes collated by GF. Revealing of majority opinions prior to the final meeting enabled experts to reflect on group thinking with potential disagreement amongst experts during Round 2 seen, although, consensus was the ultimate aim. Round 2 data was available during the meeting to aid exploration of stakeholder perspectives. In this final round sharing of opinions was promoted, and a discussion facilitated clarification of any misunderstandings [41]. Chairing of a meeting promotes encouragement of those attending and avoiding dominance of voices [42]. An effort was made to have everyone contribute to the conversations.

Consensus criteria, set before round 3, for identification of the top 4 themes from responses to questions 1 and 2 was set at 75%, and for question 4, the top single method of measuring was set at 75% of those attending the meeting [43] with opinions collected using the Zoom polling tool. Zoom possess a number of research tools and the polling tool allows the chair to collect anonymous responses to questions in real-time and feed this back to participants.

## Results

The sample size of 16 participants recruited from an email sent to 350 on a mailing list. True response rate in relation to inclusion and exclusion criteria could not be calculated. Table 3 shows the demography of the group in this study.

### Qualitative round 1

Research questions described earlier were focused on relevance to study aims and data was prepared on text document for authors to familiarise themselves with the data set. Authors CJ

**Table 3. Age and roles of participants.** SBE- Simulation based education.

| SBE role | Age |
| --- | --- |
| Outreach simulation co-ordinator | 33 |
| Clinical Teaching Fellow | 26 |
| Simulation Medical Lead | 37 |
| Programme Lead for Simulation & TEL for HEE southeast | 48 |
| Paediatric Simulation fellow at University Hospitals Bristol and Weston NHS Foundation Trust | 33 |
| Nurse Education (university) | 46 |
| Simulation Facilitator | 33 |
| Primary Care Simulation Fellow | 38 |
| Clinical Skills Tutor to final Year Medical Students | 53 |
| Resuscitation & Simulation Officer | 28 |
| GP tutor | 33 |
| Previous clinical teaching fellow | 32 |
| Clinical teaching fellow | 30 |
| Senior Technical Instructor | 44 |
| Simulation and Clinical Skills Technician | 46 |

and GW created a coding framework that represented the key codes in an inductive process and was both conducted independently and intercoder agreement assessed consistency of coding. Relationships between codes and questions are depicted in Fig 4 and this was expected with the use of questions exploring measurements and instruments.

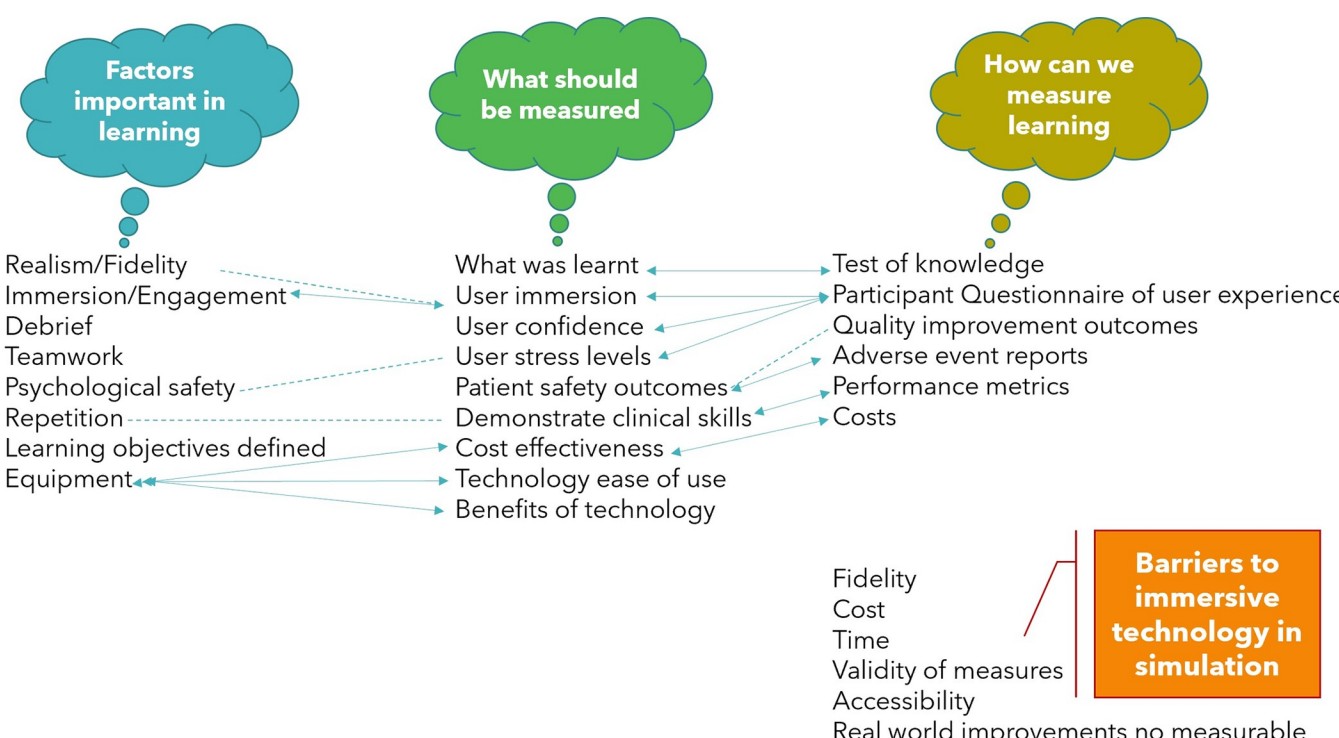

**Fig 4. Thematic map with the 4 questions abbreviated.** Arrows indicate a 2-way link of response between questions. Dotted line indicates association of responses. Red line represents barriers.

**Table 4. Descriptive results for question 1 with mean ranking.**

| Themes | Engagement Immersion | Psychological safety | Debrief | Realism/Fidelity | Teamwork | Repetition | Learning objectives defined |
|---|---|---|---|---|---|---|---|
| Valid data | 13 | 13 | 13 | 13 | 13 | 13 | 13 |
| Missing data | 0 | 0 | 0 | 0 | 0 | 0 | 0 |
| Mean | 2.76 | 3.23 | 2.85 | 3.39 | 5.62 | 5.00 | 4.15 |
| Upper quartile | 4 | 5 | 4 | 5 | 6 | 6 | 6 |
| Median | 3 | 3 | 3 | 3 | 6 | 5 | 5 |
| Lower quartile | 2 | 2 | 2 | 2 | 6 | 5 | 2 |
| IQR | 2 | 3 | 2 | 3 | 0 | 1 | 4 |

The dotted lines in Fig 4 indicate that there are patterns to the responses, whereby the codes related to one theme. Tables 4–7 present the summary of Delphi round 2 with key themes of the survey data from all 16 participants perspectives.

## Quantitative round 2

Thirteen panelists from initial 16 recruited (81%) responded to round 2.

The 4 questions were individually analysed using mean rank and Friedman's test for strength of agreement between panelists. The top ranked responses are described in relation to each question in the survey with degrees of agreement stated.

Fig 5. panel A-D are boxplots with medians and IQR for each question that give an overview of the distribution of responses.

## Question 1- List up to 5 factors that you think are important in learning in simulation?

Top four ranked responses in order: Engagement/immersion (mean rank 2.76), Debrief (mean rank 2.85), psychological safety (mean rank 3.23), and realism/fidelity (mean rank 3.39). Table 4 summarises the descriptive data for question 1.

Friedman's test indicated rank patterns ($p < 0.001$) with a degree of difference in ranking. Weighted Kappa 0.21 ($p<0.001$) indicating fair agreement in rankings between participants.

## Question 2- List what you think should be measured

Top four ranked responses in order: What was learnt (mean rank 2.54), user confidence (mean rank 3.15), user immersion (mean rank 3.69), and patient safety (mean rank 3.92). Table 5 summarises the descriptive data for question 2.

Friedman's test indicated rank patterns ($p < 0.001$) with a degree of difference in ranking. Weighted Kappa 0.22 ($p = 0.0003$) indicating a fair agreement in rankings between participants.

## Question 3- What measurement methods could you use?

Top four ranked responses in order: Participant questionnaire of user experience (mean rank 1.77), quality improvement outcomes (mean rank 2.69), performance metrics (mean rank 2.92), and test of knowledge (mean rank 3.31). Table 6 summarises the descriptive data for question 3.

Friedman's test indicated rank patterns ($p < 0.001$) with a degree of difference in ranking. Weighted Kappa 0.22 ($p < 0.001$) indicating a fair agreement in rankings between participants.

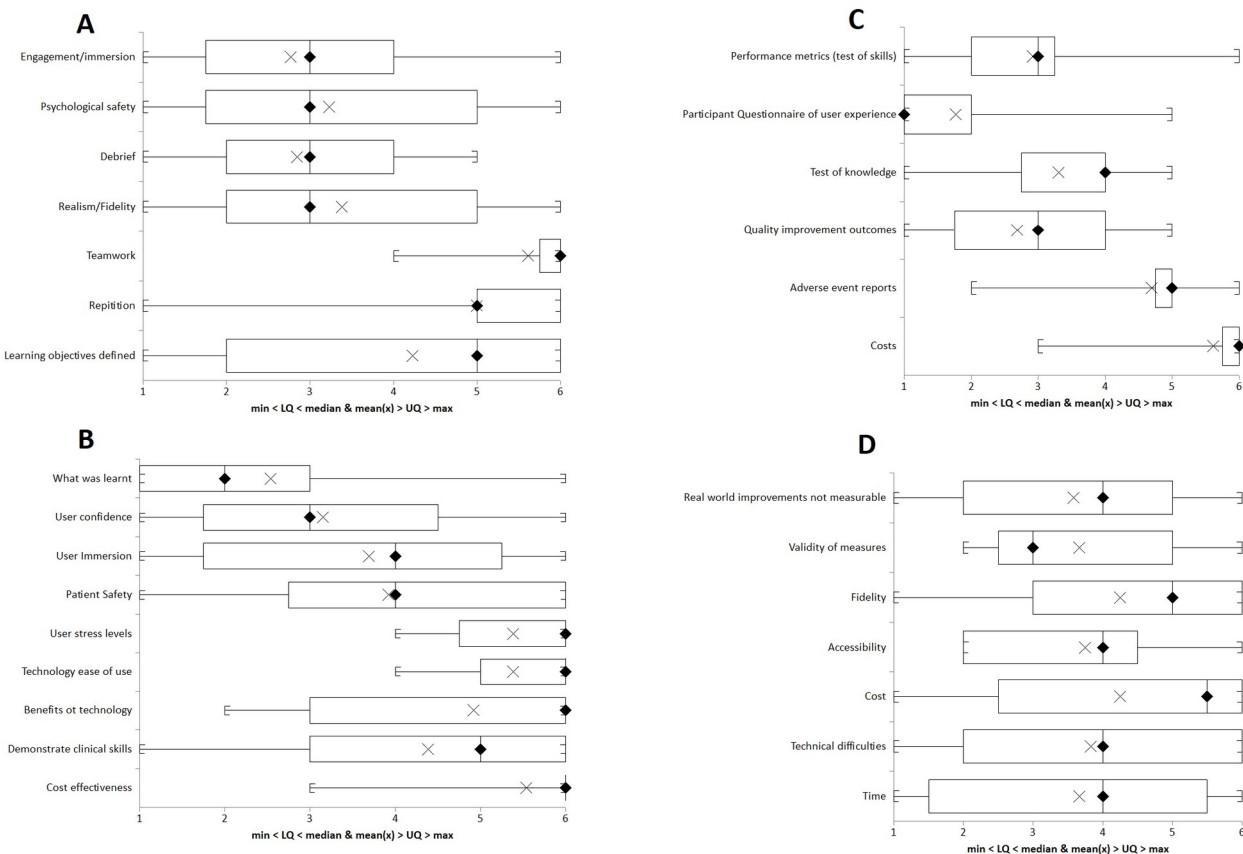

**Fig 5. Boxplot of theme rankings.** A- Question 1, B-Question 2, C-Question 3, and D- Question 4. Key, LQ-lower quartile, UQ- Upper quartile, cross marks mean, rhombus marks median.

## Question 4- List the barriers you consider in "measuring immersive technology" in simulation and medical education

Top four ranked responses in order: Real world improvements not measurable (mean rank 3.58), validity of measures (mean rank 3.67), cost (mean rank 3.67), and technical difficulties (mean rank 3.83). Table 7 summarises the descriptive data for question 4.

Friedman's test indicated no significant rank patterns ($p = 0.97$).

**Table 5. Descriptive results for question 2 with mean ranking.**

| Themes | What was learnt | User confidence | User Immersion | Cost effectiveness | Demonstrate clinical skills | Benefits of technology | Technology ease of use | User stress levels | Patient Safety |
|---|---|---|---|---|---|---|---|---|---|
| Valid data | 13 | 13 | 13 | 13 | 13 | 13 | 13 | 13 | 13 |
| Missing data | 0 | 0 | 0 | 0 | 0 | 0 | 0 | 0 | 0 |
| Mean | 2.54 | 3.15 | 3.69 | 5.54 | 4.38 | 4.92 | 5.38 | 5.38 | 3.92 |
| Upper quartile | 2 | 4 | 5 | 6 | 6 | 6 | 6 | 6 | 6 |
| Median | 2 | 3 | 4 | 6 | 5 | 6 | 6 | 6 | 4 |
| Lower quartile | 1 | 2 | 2 | 6 | 3 | 3 | 5 | 5 | 3 |
| Interquartile range | 1 | 2 | 3 | 0 | 3 | 3 | 1 | 1 | 3 |

**Table 6. Descriptive results for question 3 with mean ranking.**

| Themes | Performance metrics (test of skills) | Participant questionnaire of user experience | Test of Knowledge | Quality improvement outcomes | Adverse event reports | Costs |
|---|---|---|---|---|---|---|
| Valid data | 13 | 13 | 13 | 13 | 13 | 13 |
| Missing data | 0 | 0 | 0 | 0 | 0 | 0 |
| Mean | 2.92 | 1.77 | 3.31 | 2.69 | 4.69 | 5.62 |
| Upper quartile | 3 | 2 | 4 | 4 | 5 | 6 |
| Median | 3 | 1 | 4 | 3 | 5 | 6 |
| Lower quartile | 2 | 1 | 3 | 2 | 5 | 6 |
| Interquartile range | 1 | 1 | 1 | 2 | 0 | 0 |

## Delphi round 3

All participants of both rounds 1 and 2 were invited to join a virtual meeting hosted on Zoom. Five participants (36%) attended: 2 doctors, 2 nurses, and 1 simulation technician. There was a short presentation with a brief recap on literature surrounding immersive technology in medical education and each question was discussed in turn with results from round 2 available to prompt discussion. Table 8 shows the consensus reached for each of the 4 questions. The lack of spread to data in question 4 meant it was not appropriate for a consensus as each held equal weight.

Field notes taken during the final meeting created discussion points with each question presented. Although, no further thematic analysis was undertaken, selected quotations highlighted the expert opinion on the way we might learn from immersive technology enhanced learning (TEL) and how we might measure it. Complete field notes are available in supplementary material.

Participant J (doctor) in response to question 1.

*"I think, fidelity is high up–with enjoyment and immersion, then debrief"*

Following this statement, participant DM (simulation technician) echoed J thoughts.

*"engagement is instantaneous feedback about simulation. . .then can consolidate learning with debrief"*

Participant SK (doctor) reflecting on question 2.

*". . .simulation as a concept can be a challenge. . .cost effectiveness is hard to measure but is very important to collect data on"*

**Table 7. Descriptive results for question 4 with mean ranking.**

| Themes | Real world improvements not measurable | Validity of measures | Fidelity | Accessibility | Cost | Technical difficulties | Cost |
|---|---|---|---|---|---|---|---|
| Valid data | 13 | 13 | 13 | 13 | 13 | 13 | 13 |
| Missing data | 0 | 0 | 0 | 0 | 0 | 0 | 0 |
| Mean | 3.58 | 3.67 | 4.25 | 3.75 | 4.25 | 3.83 | 3.67 |
| Upper quartile | 5 | 5 | 6 | 5 | 6 | 6 | 5 |
| Median | 4 | 3 | 5 | 4 | 5 | 4 | 4 |
| Lower quartile | 2 | 2 | 3 | 2 | 2 | 2 | 1 |
| Interquartile range | 3 | 2 | 3 | 2 | 3 | 4 | 4 |

**Table 8. Final round consensus for each question.**

| Question | Consensus |
|---|---|
| List up to 5 factors that you think are important in learning in simulation? | 100% agreement with:<br>Engagement/Immersion<br>Psychological safety<br>Debrief<br>Realism/Fidelity |
| List what you think should be measured | 60% agreement with:<br>What was learnt<br>User confidence<br>User Immersion<br>Patient Safety<br>100% agreement if Cost effectiveness is added to above. |
| What measurement methods could you use? | 100% agreement with:<br>Participant questionnaire of user experience |
| List the barriers you consider in "measuring immersive technology" in simulation and medical education | No consensus reached |

Following this participant J responded to SK in question 2.

"*This tech is novel and is useful compared to standard methods–so immersion and testing how useful the technology is important*"

Expert consensus was quickly reached with question 3 as acceptability on measuring user experience was the most important, although it was acknowledged that this might not capture all relevant factors.

"*yes and no–good as it is easy to ask what they think–but whether the tech has made a difference after the session could be measured... difficult to measure maybe best measure is quality improvement and adverse events*"

## Discussion

Key aspects of relevant domains on measuring user experience of immersive technology in simulation were identified using the modified Delphi method in this study, with a consistency of responses evident across a heterogeneous groups of experts. It was agreed that a measure ideally would collect information on: what was learnt, the degree of immersion experienced, fidelity provided, debrief, psychological safety and patient safety. Additionally, a technological assessment should include a cost-effectiveness evaluation. There was 100% consensus in round 3 that a psychometric measure of user experience using a participant questionnaire would be the most suitable format to explore use of immersive technology in SBE.

There are numerous measures, subjective and objective, in existence for assessing a particular immersive technology in a certain situation [44]. These are often created for the purpose of the individual study in which they are used, without any argument for their validity [13]. Furthermore, there is a risk that instruments used to collect data become obsolete as emergent technology surpasses existent training methods. An international Delphi study was conducted to ensure researchers shared definitions of and terminology about instrument properties [45]. This study clarified what was meant by key measurement properties: reliability, validity, responsiveness and interpretability. These properties need to be considered when any outcome

measurement is tested. The findings of a robust literature review supported by this Delphi study have led to a conceptual model of how the utility of immersive technology in healthcare education can be measured, indeed what to measure. This will inform development of a new evidence-based measure, which can then undergo field testing with various technologies and settings, that will enable the properties to be tested and the measure refined. The process of validity is not a single task, but testing hypotheses in a continuous process to see whether scores are consistent with the model intended [46].

Although, various methods of Delphi are possible, a qualitative first round is useful as it allows experts to contribute ideas beyond current established or published knowledge. Reliability and validity of the Delphi study may be improved if the initial experts create the items for consensus reaching [47]. In this study, a relevant systematic review assisted in guiding development of questions to ask the experts.

Nasa *et al.* [50] described significant variation in how Delphi studies are conducted and reported, which can lead to uncertainty to the conclusions. Using a 9-point qualitative evaluation for a Delphi quality assessment, this study would score 8 out of 9, missing a full score only as group stability could not be evidenced. Transparency of reporting in this study is consistent with all areas in the Conducting and Reporting Delphi Studies (CREDES) checklist [39] and seen in supplementary material.

Sixteen participated in round 1 of a heterogenous panel and although, no minimum number for Delphi method exists, 15 has been suggested as minimum group size [48]. The threshold for consensus was achieved for all measure related questions, which was set *a priori*. The number of rounds of Delphi is dependent on the method and the intended outcomes. In a another modified-Delphi process, exploring learning and assessment in healthcare education, 2 rounds were used to create a consensus [49]. Delphi response rate ideally should not fall below 70% on subsequent rounds [29]: this standard was achieved in this study. Quantitative analysis was not extensive in this study due to methodology to summarise items with thematic analysis. This can be considered as a limitation to the study as Kappa values indicated fair agreement only and subsequent rounds were not conducted to seek convergence. Similarly statistical stability could not be tested for the strength of the item responses [50]. Final consensus criteria were set by a third round as an in-person virtual meeting, whereby descriptive statistics of mean ranks complemented by visual box plots to indicate spread, and a thematic map, were available to support the discussion. Thematic analysis may introduce bias, potentially inherent in individual's interpretations [51]. The primary investigator had a medical educational background, and his and the co-author's reflexive practice, and being conscious of this in the analysis, aimed to minimise impact of bias of personal experiences. Any measure developed from this process should be tested in different settings, which will reassure with regard to its validity.

Further limitation to round 3 of the study was the lower sample size of the final round and although the sample represented a range of professionals from SBE it will bias the final rankings. The triangulation of research from earlier studies aims to reduce this bias. Additionally, researchers opted for a video conferencing platfrom over a physical meeting. There are several benefits for using an online method, however, there may be a difference in interactions and rapport. Interactions in these settings have been reported as positive with visual and nonvisual cues available to participants [52], yet a proportion still favour a physical meeting [40].

The Delphi is not a psychometric instrument in itself [15] but a practical method for gauging a group-based judgment, more useful for complex concepts and decisions. It can be argued that there is not a preferred Delphi method [53], however, the process of a groupthink interaction generates ideas and aids the conceptual framework that this study addressed.

## Conclusion

This study explained the process of sourcing opinion from a varied cohort of experts to determine how the experience of a learner in healthcare practice using immersive technology might be measured. Participants agreed to a self-reported measure on user; depth of immersion; fidelity of experience; psychological safety of a scenario; and aspects of reflection. Synthesising these results in to a multidomain instrument requires a systematic approach to testing in future research. There is variation between educationalists in what is regarded as important in learning from this technology, however, it is important to establish relevant experts' opinions before designing a user experience outcome measure, and this study highlighted a consensus on the topic.

## Supporting information

**S1 Checklist. CREDES Checklist: Recommendations for the Conducting and REporting of DElphi Studies (CREDES).**
(DOCX)

**S2 Checklist. *PLOS ONE* clinical studies checklist.**
(DOCX)

**S1 Text. Delphi round 1 results anonymised.**
(XLSX)

**S2 Text. Delphi round 2 results anonymised.**
(RTF)

**S3 Text. Delphi round 3 field notes anonymised.**
(DOCX)

## Author Contributions

**Conceptualization:** Chris Jacobs, Michael Williams.

**Data curation:** Chris Jacobs, Georgia Foote.

**Formal analysis:** Chris Jacobs, Georgia Foote, Michael Williams.

**Funding acquisition:** Chris Jacobs.

**Investigation:** Chris Jacobs.

**Methodology:** Chris Jacobs, Michael Williams.

**Project administration:** Chris Jacobs, Georgia Foote.

**Resources:** Chris Jacobs.

**Software:** Chris Jacobs.

**Supervision:** Chris Jacobs, Michael Williams.

**Validation:** Chris Jacobs.

**Visualization:** Chris Jacobs.

**Writing – original draft:** Chris Jacobs.

**Writing – review & editing:** Chris Jacobs, Michael Williams.

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
