## [Decision Letter · Decision Letter 0]

15 May 2023

PONE-D-22-26422Evaluating user experience with immersive technology in simulation-based education: a modified Delphi study with qualitative analysisPLOS ONE

Dear Dr. Jacobs,

Thank you for submitting your manuscript to PLOS ONE. After careful consideration, we feel that it has merit but does not fully meet PLOS ONE’s publication criteria as it currently stands. Therefore, we invite you to submit a revised version of the manuscript that addresses the points raised during the review process.

We look forward to receiving your revised manuscript.

Kind regards,

Soham Bandyopadhyay

Academic Editor

PLOS ONE

“CJ received a grant from Simulation West Network as part of Health Education England 2022 funding. No number is associated with grant and no sponsor involvement took place.”

4. We note you have included a table to which you do not refer in the text of your manuscript. Please ensure that you refer to Table 8 in your text; if accepted, production will need this reference to link the reader to the Table.

5. We note that Figures 1, 2 and 3 in your submission contain copyrighted images. All PLOS content is published under the Creative Commons Attribution License (CC BY 4.0), which means that the manuscript, images, and Supporting Information files will be freely available online, and any third party is permitted to access, download, copy, distribute, and use these materials in any way, even commercially, with proper attribution. For more information, see our copyright guidelines: http://journals.plos.org/plosone/s/licenses-and-copyright.

a. You may seek permission from the original copyright holder of Figures 1, 2 and 3 to publish the content specifically under the CC BY 4.0 license.

b.If you are unable to obtain permission from the original copyright holder to publish these figures under the CC BY 4.0 license or if the copyright holder’s requirements are incompatible with the CC BY 4.0 license, please either i) remove the figure or ii) supply a replacement figure that complies with the CC BY 4.0 license. Please check copyright information on all replacement figures and update the figure caption with source information. If applicable, please specify in the figure caption text when a figure is similar but not identical to the original image and is therefore for illustrative purposes only.

6. Please upload a new copy of Figure 4 as the detail is not clear. Please follow the link for more information: https://blogs.plos.org/plos/2019/06/looking-good-tips-for-creating-your-plos-figures-graphics/" https://blogs.plos.org/plos/2019/06/looking-good-tips-for-creating-your-plos-figures-graphics/

**Comments to the Author**

1. Is the manuscript technically sound, and do the data support the conclusions?

Reviewer #1: Partly

Reviewer #2: Yes

Reviewer #3: Yes

2. Has the statistical analysis been performed appropriately and rigorously? 

Reviewer #1: I Don't Know

Reviewer #2: Yes

Reviewer #3: Yes

3. Have the authors made all data underlying the findings in their manuscript fully available?

Reviewer #1: No

Reviewer #2: Yes

Reviewer #3: Yes

4. Is the manuscript presented in an intelligible fashion and written in standard English?

Reviewer #1: Yes

Reviewer #2: Yes

Reviewer #3: Yes

5. Review Comments to the Author

Reviewer #1: The details of the study are not shown so it is difficult to accurately assess the work. The Delphi method modification seems to be applied adequately. It is not clear how the Delphi method is particularly adapted to immersive technologies. I would expect some reference to user performance or usability in relation to the immersive to potentially draw contrasts between the user experience and preferences. If this was done, it is not evident in the manuscript.

Reviewer #2: The research aims to establish a consensus of items and domains that different simulation experts would include in a measure for immersive technology use in simulation-based medical education. The study addresses the lack of validated measures that capture user experience to inform the technology utility. The study concludes that there is varied opinion on what should be prioritized in measuring the experience in simulation practice. However, the article does not discuss the originality of the topic or how it adds to the subject area compared to previous research. More related studies can be highlighted in the introduction section. The study involved sixteen simulation experts who participated in the study, but in the quantitative round 2 participated only 13 and round 3 just five - a rather weak sample. Discuss how the limited number of participants from round 3 influence the study's conclusions. Fig 4, hard to read, please increase the font size.

Regards

Reviewer #3: The author is an excellent talent and has enough research on the professional field.The article has a good representation, a lot of interpretation of the topic of the article. The data given by the author is also quite in-depth and clear. It shows that the author has done a lot of research.

6. PLOS authors have the option to publish the peer review history of their article (what does this mean?). If published, this will include your full peer review and any attached files.

Reviewer #1: No

Reviewer #2: No

Reviewer #3: No

---

## [Author Response · Author response to Decision Letter 0]

22 Jun 2023

Please find reviewers comments document in the submission

---

## [Editor Report · Decision Letter 1]

11 Jul 2023

Evaluating user experience with immersive technology in simulation-based education: a modified Delphi study with qualitative analysis

PONE-D-22-26422R1

Dear Dr. Jacobs

We’re pleased to inform you that your manuscript has been judged scientifically suitable for publication and will be formally accepted for publication once it meets all outstanding technical requirements.

Kind regards,

Soham Bandyopadhyay

Academic Editor

PLOS ONE

---

## [Editor Report · Acceptance letter]

25 Jul 2023

PONE-D-22-26422R1 

Evaluating user experience with immersive technology in simulation-based education: a modified Delphi study with qualitative analysis 

Dear Dr. Jacobs:

I'm pleased to inform you that your manuscript has been deemed suitable for publication in PLOS ONE. Congratulations! Your manuscript is now with our production department. 

Kind regards, 

on behalf of

Dr. Soham Bandyopadhyay 

Academic Editor

PLOS ONE